# Integrated Analysis of miRNA-mRNA Network Reveals Different Regulatory Patterns in the Endometrium of Meishan and Duroc Sows during Mid-Late Gestation

**DOI:** 10.3390/ani10030420

**Published:** 2020-03-03

**Authors:** Kaijie Yang, Jue Wang, Kejun Wang, Yabiao Luo, Qiguo Tang, Ximing Liu, Meiying Fang

**Affiliations:** 1Department of Animal Genetics and Breeding, National Engineering Laboratory for Animal Breeding, MOA Laboratory of Animal Genetics and Breeding, Beijing Key Laboratory for Animal Genetic Improvement, College of Animal Science and Technology, China Agricultural University, Beijing 100193, China; kjyang2014@163.com (K.Y.); tctcttc@hotmail.com (J.W.); wangkejun@163.com (K.W.); mothluo@foxmail.com (Y.L.); tango@cau.edu.cn (Q.T.); Lximing2018@163.com (X.L.); 2College of Animal Science and Veterinary Medicine, Henan Agricultural University, Zhengzhou 450002, China

**Keywords:** RNA-seq, miRNA-mRNA interaction, endometrium, Meishan and Duroc sows

## Abstract

**Simple Summary:**

Meishan pigs have a lower fetal loss rate during mid-late gestation compared to Duroc pigs. Differentially expressed mRNAs and miRNAs detected in endometrial tissue from Meishan and Duroc sows at mid-late gestation are involved in regulating hormone and oxygen levels, blood vessel development, and developmental processes affecting reproduction. In addition, ssc-miR-503 and ssc-miR-671-5p were shown to target the *EGF* and *ESR1* genes, respectively. These findings provided an important resource for studying embryonic mortality during mid-late gestation in pigs.

**Abstract:**

Embryo loss is a major factor affecting profitability in the pig industry. Embryonic mortality occurs during peri-implantation and mid-late gestation in pigs. Previous investigations have shown that the embryo loss rate in Meishan pigs is significantly lower than in commercial breeds. Most studies have focused on embryonic mortality during early gestation, but little is known about losses during mid-late gestation. In this study, we performed a transcriptome analysis of endometrial tissue in mid-late gestation sows (gestation days 49 and 72) sampled from two breeds (Meishan (MS) and Duroc (DU)) that have different embryo loss rates. We identified 411, 1113, 697, and 327 differentially expressed genes, and 14, 36, 57, and 43 differentially expressed miRNAs in four comparisons (DU49 vs. DU72, DU49 vs. MS49, DU72 vs. MS72, and MS49 vs. MS72), respectively. Subsequently; seven differentially expressed mRNAs and miRNAs were validated using qPCR. Functional analysis suggested the differentially expressed genes and miRNAs target genes mainly involved in regulation of hormone levels, blood vessel development, developmental process involved in reproduction, embryonic placenta development, and the immune system. A network analysis of potential miRNA-gene interactions revealed that differentially expressed miRNAs in Meishan pigs are involved in the response to estradiol and oxygen levels, and affect angiogenesis and blood vessel development. The binding site on ssc-miR-503 for epidermal growth factor (*EGF*) and the binding site on ssc-miR-671-5p for estrogen receptor α (*ESR1*) were identified using a dual luciferase assay. The results of this study will enable further exploration of miRNA-mRNA interactions important in pig pregnancy and will help to uncover molecular mechanisms affecting embryonic mortality in pigs during mid-late gestation.

## 1. Introduction

Litter size, which depends on ovulation rate, fertilization rate, and conceptus survival to term, is one of the most important economic traits affecting production efficiency in pig the industry [1]. Although fertility in the sow is directly linked to the ovulation rate, the main barrier to increasing litter size in pigs is prenatal mortality [2]. The Chinese Meishan breed is one of the most prolific, bearing three to five piglets more per litter than European and North American commercial breeds despite having a similar ovulation rate [3]. In contrast, the Duroc is considered to be a low-prolificacy breed. Meishan embryos develop more slowly and produce less estrogen, resulting in increased embryo survival [4,5]. Meishan pigs also possess higher uterine capacity and placental efficiency than other commercial breeds, due to greater vascular density in the developing placenta and endometrium [6,7].

Between 20–45% of embryos are lost in pig pregnancy, which is a major challenge to the commercial swine industry [8]. In pigs, embryonic losses occur during two critical developmental stages; 20–30% occur during the peri-implantation (day 10–30 of gestation (GD10–30)), and an additional 10–15% of embryos are lost at mid-late gestation (GD50–90) [1,8]. Although investigators have identified genetics [9], uterine capacity [10], and angiogenesis at the maternal-fetal interface [11] as important factors, the underlying reasons responsible for spontaneous conceptus loss are still not well understood. After GD30, embryo losses in swine are thought to result from intrauterine crowding caused by limited uterine capacity [12].

miRNAs are a class of non-coding RNAs, approximately 22 nucleotides in length, which negatively regulate the translation of target genes by binding to complementary sequences in the 3′UTR (untranslated region) [13,14]. In mammals, miRNAs are predicted to control the activity of about 50% of all protein-coding genes and regulate gene expression in every cellular pathway [15]. In pigs, miRNA-mRNA interactions play an important role in pregnancy. Two important miRNAs, miR-26a and miR-125b, have been identified in extracellular vesicles isolated from uterine luminal flushings during early pregnancy. miR-125b regulates the expression of genes involved in embryo development and implantation [16]. In addition, miR-181a and miR-181c are thought to have important roles in embryo implantation and placentation in pigs by targeting *SPP1* (secreted phosphoprotein 1), *ESR1* (estrogen receptor 1), and *ITGB3* (integrin beta3) [17]. Finally, miR-130b appears to facilitate the expression of *HPSE* (heparanase), which changes placental folding and increases placental efficiency by suppressing the expression of *PPARG* (peroxisome proliferator activated receptor gamma) [18].

Previous studies show that Meishan pigs have less embryo loss than other commercial breeds, not only during early gestation but also mid-late gestation [4,5,19]. The studies conducted using microarrays and RNA-seq to identify and characterize the genes and miRNAs expressed in the uterine endometrium during early gestation [20,21,22,23] are an important step toward understanding the complex processes that govern the establishment and maintenance of pregnancy. However, little is known about genes and miRNAs expressed in the uterine endometrium during mid-late gestation.

Here, we used RNA-sequencing (RNA-seq and miRNA-seq) to compare the uterine endometrium transcriptomes from Chinese Meishan and Duroc sows during mid-late gestation. The data and our analysis may provide deeper insight into the molecular mechanisms affecting embryonic mortality in pigs.

## 2. Materials and Methods

### 2.1. Animal Sample Collection and RNA Extraction

Six unrelated Meishan pigs (no common grandparents) and six unrelated Duroc pigs were selected at two different stages of gestation (GD49 and GD72). The sows were divided into four experimental groups, each of which contained three sows of the same breed (MS49, MS72, DU49, DU72). The pigs were raised in identical conditions by the Shanghai Zhu Zhuang Yuan Company (Shanghai, China). All pigs had previously given birth to three litters and were in their fourth pregnancy. Pigs at the same day of gestation were slaughtered in the same week. After slaughter, endometrial samples were collected from each pig and stored until use at −80 °C or in liquid nitrogen. Animal treatment and sample collection were approved by the Animal Welfare Committee of the State Key Laboratory of Agricultural Biotechnology, China Agricultural University (approval No. XK257), and in accordance with the Regulations on Administration of Affairs Concerning Experimental Animals, revised June 2004 (Ministry of Science and Technology).

Total RNA was extracted from endometrial tissues using TRIzol^®^ Reagent (Invitrogen, San Diego, CA, USA), using the manufacturer’s protocol. RNA degradation and contamination were evaluated on a 1% agarose gel. A Qubit 2.0 Fluorometer and Qubit RNA analysis kit (Life Technologies, Carlsbad, CA, USA) were used to measure RNA concentration. RNA purity and integrity were checked using the RNA Nano 6000 Assay Kit for the Bioanalyzer 2100 system (Agilent Technologies, Foster, CA, USA) and a NanoPhotometer^®^ spectrophotometer (IMPLEN, Westlake Village, CA, USA), respectively.

### 2.2. Library Preparation for mRNA Sequencing and Data Analysis

A quantity of 3 µg RNA from each sample was used for sample preparation. Sequencing libraries were constructed according to the manufacturer’s recommendations using the NEBNext^®^ Ultra^TM^ RNA Library Prep Kit for Illumina^®^ (NEB, Ipswich, MA, USA), and index adapters were added to each sample. Briefly, poly-T oligo-attached magnetic beads were used to purify mRNA from total RNA. Divalent cations were used to fragment RNA at high temperature in NEBNext First Strand Synthesis Reaction Buffer (5×). The first cDNA strand was synthesized using random hexamer primers and M-MuLV Reverse Transcriptase (RNase H-). The second strand was synthesized using DNA polymerase I and RNase H. Overhangs were converted into blunt ends by the exonuclease/polymerase activity of PolI. After the 3′ ends of the DNA fragments were adenylated, they were ligated to EBNext Adaptors. To enrich for cDNA fragments around 250–300 bp, library fragments were purified using the AMPure XP system (Beckman Coulter, Beverly, MA, USA). The size-selected adaptor-ligated cDNA was incubated with 3 μL USER Enzyme (NEB, Ipswich, MA, USA) at 37 °C for 15 min, followed by 5 min at 95 °C. PCR was then conducted with Phusion High-Fidelity DNA polymerase, Universal PCR primers, and Index (X) Primer. Finally, the PCR products were purified using the AMPure XP system, and library quality was evaluated with the Agilent Bioanalyzer 2100 system. A HiSeq 2000 platform (Illumina, San Diego, CA, USA) was used to generate 150 bp paired-end reads. Raw reads in FASTQ format were initially processed using custom Perl scripts. During this step, clean data were obtained by removing reads with adapter sequences, poly-N sequences, or low-quality scores. Q20, Q30, and GC content were calculated to evaluate the quality of the clean data, and only high-quality clean data were used in subsequent analyses. Reads were mapped to the pig reference genome (Sus scrofa 11.1, Ensembl, ftp://ftp.ensembl.org/pub/release-95/fasta/sus_scrofa/dna/) with HiSAT2 (http://ccb.jhu.edu/software/hisat2/index.shtml) [24]. The number of reads for each gene was determined by featureCounts v1.5.0-p3 (http://subread.sourceforge.net) [25], and FPKM (fragments per kilobase of transcript per million fragments mapped) for each gene was calculated according to gene length and read count. Differences in mRNA expression between Meishan and Duroc sows were analyzed using the DESeq2 R package (1.16.1) (http://www.bioconductor.org/packages/release/bioc/html/DESeq2.html) [26]. mRNAs with an adjusted *p*-value < 0.05 were classified as differentially expressed genes (DEGs).

### 2.3. Library Preparation for MicroRNA Sequencing and Data Analysis

A quantity of 3 µg RNA from each sample was used to construct a small RNA library. Sequencing libraries were generated according to the manufacturer’s protocol accompanying the NEBNext^®^ Multiplex Small RNA Library Prep Set for Illumina^®^ (NEB, Ipswich, MA, USA). Index adapters were added to each sample. Briefly, NEB 3′ SR Adapters were directly ligated to the 3′ ends of the RNA. After the 3′ ligation reaction, SR RT Primers were hybridized to the free 3′ SR adaptors (i.e., unligated adaptors remaining after the ligation reaction) to transform the single strand DNA adaptors into double-strand DNA molecules. The first cDNA strand was then synthesized using M-MuLV reverse transcriptase (RNase H -). PCR amplification was conducted using LongAmp Taq 2X Master Mix, Index (X) Primer, and SR Primer from Illumina. After purification on an 8% polyacrylamide gel (100 V, 80 min), DNA fragments between 140 and160 bp (the length of small non-coding RNA plus 3′ and 5′ adaptors) were recovered and dissolved in 8 μL elution buffer. Finally, the quality of the library was evaluated with an Agilent Bioanalyzer 2100 system using DNA High Sensitivity Chips. Following the manufacturer’s instructions, a TruSeq SR Cluster Kit v3-cBot-HS (Illumina, NEB, Ipswich, MA, USA) was used to cluster the index-coded samples on a cBot Cluster Generation System (Illumina, NEB, Ipswich, MA, USA). After clustering, the library was sequenced on the Illumina Hiseq 2500/2000 platform and 50 bp single-end reads were generated. Raw reads were collected in FASTQ format and initially processed using custom Perl and Python scripts. In this step, clean data were generated by removing reads with low-quality scores and reads containing poly-n, 5′ adapters, poly A or T or G or C, or that did not contain 3′ adapters or insert tags. Q20, Q30, and GC content were calculated to evaluate clean data quality. Clean reads, 18–35 nucleotides in length, were selected for all subsequent analyses. Bowtie [27] was used to align small RNAs to the porcine reference genome (Sus scrofa 11.1), and Bedtools (https://bedtools.readthedocs.io/) was used to identify known miRNAs by matching them to entries in miRBase20.0 (http://www.mirbase.org/). After excluding matching reads, the remaining reads were reanalyzed to predict novel miRNAs using miRDeep2 [28]. miRNA expression was estimated by RPM (reads per million total reads). Differential expression of miRNA between samples was analyzed with the DEGseq R package (1.8.3) (http://bioinfo.au.tsinghua.edu.cn/software/degseq) [29]. miRNAs with significantly different levels of expression (*p*-value < 0.05) were classified as differentially expressed (DE) miRNAs.

### 2.4. Gene Ontology and Kyoto Encyclopedia of Genes and Genomes Enrichment Analyses

Metascape (http://metascape.org) [30] was used to conduct gene ontology (GO) enrichment analysis of the DEGs and target genes of DE miRNAs. Pig gene ensemble IDs were first converted into human gene IDs by the Ensembl BioMart tool (http://www.ensseml.org/biomart/martview) because the functional annotation for the pig is incomplete. The corresponding human gene ensemble IDs were then submitted to the Metascape database for functional annotation. GO items with adjusted *p*-values less than 0.01 were considered to be significantly enriched by DEGs or target genes of DE miRNAs. We used KOBAS V3.0 (http://kobas.cbi.pku.edu.cn) [31] to test the statistical significance of the enrichment of DEGs or target genes of DE miRNAs in Kyoto Encyclopedia of Genes and Genomes (KEGG) pathways. Pathways with *p*-values lower than 0.05 were regarded as significantly enriched.

### 2.5. Integrated miRNA-mRNA Analysis

To construct the miRNA-target gene network, BLASTN (https://blast.ncbi.nlm.nih.gov) was first used to recognize and remove pre-microRNAs based on high levels of similarity. Then, target relationships with miRNAs were predicted by miRanda [32], requiring an alignment score N > 140 and energy < −10 kcal/mol. Further analyses of miRNA-gene pairs were conducted based on the common miRNA-binding sites. The miRNA-mRNA interaction network was constructed and visualized using Cytoscape v3.7.2 (https://cytoscape.org) [33].

### 2.6. Quantitative Polymerase Chain Reaction

To detect DEGs and DE miRNAs, 1 g of RNA from endometrium tissue was transcribed into cDNA using a FastQuant RT Kit (with gDNase) (Tiangen Biotech Co., Ltd., Beijing, China) and a miRcute Plus miRNA First-Strand cDNA Kit (Tiangen Biotech Co., Ltd., Beijing, China), following the manufacturer’s instructions. The expression levels of seven genes and seven miRNAs were detected by quantitative PCR with a miRcute Plus miRNA qPCR Kit (Tiangen Biotech Co., Ltd., Beijing, China) and SuperReal PreMix Plus (SYBR Green) (Tiangen Biotech Co., Ltd., Beijing, China), respectively. Primer Premier 5.0 (Premier Biosoft International, Palo Alto, CA, USA) was used to design primers for qPCR of genes and miRNAs, and the primers were synthesized by SANGON biotechnology company (Beijing, China). Gene and miRNA primer sequences are listed in Appendix A. The cycling parameters used for qPCR amplification of genes were as follows: initial heat denaturation at 95 °C for 15 min; 40 cycles at 95 °C for 30 s, 60 °C for 30 s, and 72 °C for 30 s; and a final extension at 72 °C for 5 min. The cycling parameters used for qPCR amplification of miRNAs were as follows: initial heat denaturation at 95 °C for 15 min; 5 cycles at 94 °C for 20 s, 65 °C for 30 s, and 72 °C for 34 s; 40 cycles at 94 °C for 20 s and 60 °C for 34 s. A melting curve analysis was conducted to eliminate DNA contamination and confirm primer specificities. The 2^−∆∆Ct^ method with *β-actin* was used as the endogenous control for the normalization of gene expression levels. Relative miRNA expression was also normalized using the 2^−∆∆Ct^ method with the U6 small nuclear RNA as an internal standard. Forward primers of U6 were designed and synthesized by RiboBio (Guangzhou, China). Each biological replica consists of three technical replicates.

### 2.7. Cell Culture and Transfection

HEK293T cells were obtained from the Institute of Biochemistry and Cell Biology, Chinese Academy of Science, P. R. China. HEK293T cells were cultured in DMEM with 10% FBS and 1% penicillin/streptomycin. The cells were cultured at 37 °C with 5% CO_2_. All reagents for cell culture were purchased from Gibco (Langley, OK, USA) and Invitrogen (Carlsbad, CA, USA).

### 2.8. Dual Luciferase Assay

Potential binding sites for ssc-miR-503 and ssc-miR-671-5p were predicted using miRanda as described above. The target sequence for ssc-miR-503 (within the CDS of the *EGF* gene) and the target for ssc-mir-671-5p binding (within the 3′-UTR region of *ESR1*) were amplified from pig genomic DNA and cloned into the psiCHECK-2 plasmid (Promega, Madison, WI, USA) using the XhoI and NotI restriction sites downstream from the Renilla luciferase gene. The two primer pairs are shown in Appendix A. Mutant plasmids with altered putative binding sites were synthesized by BGI (Beijing), and Geneprama (Suzhou, China) synthesized ssc-miR-503 and ssc-miR-671-5p mimics and negative control mimics. Plasmids containing mutant or wild-type binding sites were co-transfected into HEK293T cells with either negative control mimics or miRNA mimics at a concentration of 100 nM using Lipofectamine^TM^ 2000 (Invitrogen, Carlsbad, CA, USA). Experiments were conducted in triplicate. Cells were collected 36 hours after transfection and luciferase activity was measured using the Dual-Glo^®^ Luciferase Assay System (Promega, Madison, WI, USA) in accordance with the manufacturer’s instructions. The data were first normalized by the ratio of Renilla luminescence (CDS or 3′UTR region) to firefly luminescence (transfection control). Relative luciferase activity was defined by calculating the ratio of the mean of the three biological replicates in each miRNA transfection group to the mean of the corresponding control siRNA transfection group.

### 2.9. Statistical Analysis

Data are shown as means ± standard deviation (SD). A one-way ANOVA was used to test for significance. Levene’s test was used to assess homogeneity of variance, followed by Student’s t-test. Data analysis was conducted using SAS version 9.0 (SAS, Cary, NC, USA). Differences were regarded as statistically significant for *p*-values < 0.05.

## 3. Results

### 3.1. Overview of Sequencing Data

Between 51.29 to 66.25 million raw reads were acquired for each sample. After quality control, 48.76 to 61.19 million clean reads remained and were aligned to the pig reference genome (Sus scrofa 11.1) (Appendix A). Approximately 90% of total reads could be mapped, and 87% mapped to unique genomic positions (Appendix A). As expected, most reads mapped to coding regions (84.13–90.84%), whereas 5.16–9.63% mapped to introns and 4.00–6.24% mapped to intergenic regions (Appendix A). Expression of 21,333 genes was detected in the pig endometrial tissues, and 17,779 genes exhibited expression in all four experimental groups. The miRNA libraries generated an average of 15.73 million raw reads. After quality control, about 97.8% of raw reads were retained for further analysis (Appendix A). Most reads were 20–24 nt, with a modal length of 22 nt. An average of 13.76 million (93.40%) of the total sRNAs were aligned to the pig reference genome (Sus scrofa 11.1) (Appendix A). 484 mature miRNAs were identified, including 355 annotated mature miRNAs from 321 precursors (Appendix A) and 129 novel mature miRNAs from 134 precursors (Appendix A).

### 3.2. Identifying Differentially Expressed mRNAs and miRNAs

Using RNA samples from two points of gestation, we performed four comparisons (DU49 vs. DU72, DU49 vs. MS49, DU72 vs. MS72 and MS49 vs. MS72). The analyses revealed, 411, 1113, 697, and 327 genes that met criteria for differential expression (adjusted *p*-value < 0.05), respectively (Table 1 and Appendix A). Several differentially expressed genes have previously been identified as important for fetal loss or litter size (Table 2).

We also identified 14, 36, 57, and 43 DE miRNAs (*p*-value < 0.05) in the four comparisons described above (Appendix A). Among these, miR-19a [50], miR-323 [51], miR-27a [51,52], miR-30a-5p [53] may be involved in fetal loss during pregnancy. Venn diagrams showing the differentially expressed genes and miRNAs from the four groups are presented in Figure 1A (mRNAs) and Figure 1B (miRNAs).

### 3.3. Functional Analysis of DEGs and DE miRNA Target Genes

GO and KEGG pathway analyses were performed to evaluate the potential functions of DEGs. The most enriched terms and pathways associated with DEGs identified in the comparison of GD49 and GD72 are shown in Figure 2. Similarly, terms and pathways identified in the comparison of Meishan and Duroc pigs are shown in Figure 3. Pregnancy-associated GO biological process (BP) terms were significantly enriched for DEGs from all four comparisons, such as “regulation of hormone levels”, “blood vessel development”, “developmental process involved in reproduction”, “embryonic placenta development”, and “immune” (Appendix A). KEGG pathways related to pregnancy were also enriched significantly in the four comparison groups, such as “steroid hormone biosynthesis”, “ovarian steroidogenesis”, and “estrogen signaling pathway” (Appendix A).

miRanda was used to predict DE miRNA target genes. A GO analysis suggests that DE miRNAs target genes primarily associated with the terms “metabolic processes”, “biological regulation”, “developmental processes”, and “immune” (Appendix A). “Rap1 signaling pathway”, “ras signaling pathway”, “ovarian steroidogenesis”, and “vascular smooth muscle contraction” were significantly enriched in the KEGG pathway analysis (Appendix A).

### 3.4. miRNA-mRNA Interaction Analysis

To characterize the regulatory roles of miRNAs in the endometrium during mid-late gestation, we used miRanda to predict potential target relationship between DE miRNAs and DEGs. We identified 3 (DU49 vs. DU72), 18 (DU49 vs. MS49), 22 (DU72 vs. MS72), and 30 (MS49 vs. MS72) potential mRNA targets for 12, 131, 250, and 129 miRNAs, respectively. The predictions were used to construct four miRNA-gene interaction networks (Figure 4, Figure 5, Figure 6 and Figure 7 and Appendix A), which show possible interactions amongst these genes and miRNAs. GO and KEGG pathway analyses were performed for DEGs in the networks. Several significantly enriched GO terms were shared by DEGs from MS49 vs. MS72 and DU72 vs. MS72, including “response to estradiol”, “response to oxygen levels”, “angiogenesis”, and “blood vessel development” (Table 3). DEGs corresponding to the GO terms are significantly enriched in the Rap1, Ras, PI3K-Akt, and FoxO signaling pathways (Table 3). Interestingly, ssc-miR-503 potentially binds *EGF* mRNA and ssc-miR-671-5p potentially binds *ESR1* mRNAs. Both genes are involved in litter size in pregnancy (Table 2). 

### 3.5. qPCR Validation for DEGs and DE miRNAs

Seven differentially expressed mRNAs (Appendix A) and seven differentially expressed microRNAs (Appendix A) were selected to validate the RNA-sequencing results. Among them, *EGF*, *ESR1*, ssc-miR-503, and ssc-miR-671-5p were selected because they appear to be involved in the determination of litter size based on the miRNA-gene interaction networks we constructed. The remaining five DE mRNAs and DE microRNAs were randomly selected from the DEGs and DE miRNAs identified in the four comparison groups. The expression patterns for all 14 RNAs were examined in Meishan and Duroc endometrium using quantitative RT-PCR. The results were consistent with the RNA-sequencing data (Figure 8A–N).

### 3.6. Validation of miRNA-mRNA Interactions

The miRNA-mRNA interaction analyses suggest that the CDS region of *EGF* contains a potential target site for ssc-miR-503, and the 3′UTR region of *ESR1* contains a potential target site for ssc-miR-671-5p. We chose these two potential target relationships to test for binding between miRNA and mRNA targets. The nucleotide sequence at the potential target sites are identical in Meishan and Duroc pigs. We conducted dual luciferase assays to detect miRNA-mRNA interactions in HEK293T cells. As shown in Figure 9A, ssc-miR-503 mimics decreased luciferase activity compared with the negative control mimics when co-transfected with the *EGF* wild-type plasmid in HEK293T cells. In contrast, co-transfection with a plasmid containing a binding site mutation or the psiCHECK-2 plasmid had no effect on activity. The luciferase activity of co-transfected ssc-miR-671-5p mimics and *ESR1* plasmid yielded similar results (Figure 9B). These data support the conclusion that ssc-miR-503 directly targets the *EGF* gene CDS region and ssc-miR-671-5p directly targets the *ESR1* gene 3′UTR region.

## 4. Discussion

Meishan pigs have lower prenatal mortality than commercial breeds, including Duroc [1,8]. The molecular mechanisms affecting embryonic mortality are still not well understood, especially during mid-late gestation. In this study, numerous DEGs and DE miRNAs were identified in comparisons of Meishan and Duroc endometrial tissue at GD49 and GD72, revealing that major temporal changes occur in the endometrium during mid-late gestation. These shifts may be involved in embryo loss rate differences between Meishan and Duroc pigs. 

In our study, the miRNA-mRNA interaction analyses show that miRNAs bind not only the 3′UTR of target genes, but also the CDS region of the target genes, which is consistent with previous research results [54]. Some DE miRNAs identified in our study play a significant role in pregnancy loss. The miR-27a polymorphism, for example, is significantly associated with pregnancy loss in humans [52]. Consistent with our findings, Wessels et al. identified miR-323 and miR-27a as potentially involved in spontaneous embryo loss in pigs [51]. Our miRNA-gene interaction analyses identified a much large number of predicted miRNA-gene pairs in the comparison of MS49 and MS72 than in DU49 vs. DU72, suggesting that Meishan sows have a greater ability to regulate the uterine environment. In addition, unlike Duroc sows at same period or Meishan sows at GD49, DE genes in the Meishan endometrium at GD72 were associated with pregnancy hormones, hypoxia, and angiogenesis (Table 3). Genes associated with these terms may participate in blood vessel development and increase the density of blood vessels in the endometrium of Meishan pigs. For example, epidermal growth factor (*EGF*) can improve angiogenesis by directly stimulating endothelial cell proliferation [55], and *EGF* expression is higher at MS72 than at DU72. Studies have revealed that Meishan pigs have higher placental efficiency near the end of gestation because of dense vascularization, in comparison to commercial breeds [6]. Using the luciferase reporter system, we identified binding site for ssc-miR-503 in the *EGF* CDS region (Figure 9A). The relatively low expression of ssc-miR-503 in Meishan pigs at GD72 compared with Duroc pigs (Figure 8L) may enable expression of *EGF* (Figure 8C) and facilitate blood vessel development in the endometrium.

Angiogenesis occurs at the maternal-fetal interface and plays a crucial role in the development of growing conceptuses after embryo implantation [1]. In our study, GO analysis of DEGs between GD49 and GD72 in both Meishan and Duroc revealed significantly enriched GO terms involved in blood vessel development (Figure 2 and Appendix A). The only gene found in common in the two comparisons was *APLN*, encoding apelin. Apelin is an endogenous peptide that functions as the ligand of the G protein-coupled receptor APJ, which may be involved in many physiological processes including angiogenesis [56]. Apelin plays a role in regulating the hemodynamics of pregnancy [57]. Increased expression of Apelin promotes the growth, migration, and angiogenesis of endothelial progenitor cells under conditions of hypoxia [58]. Using quantitative PCR, we found a higher level of *APLN* expression in GD72 than GD49 (Figure 8A). These results suggest that *APLN* plays a vital role in blood vessel development during mid-late gestation in the endometrium to meet increasing fetal demands for oxygen and nutrients. The top 20 most enriched GO terms for genes identified in the comparison between Meishan and Duroc at GD72 include several involved in blood vessel development. This is not the case for the corresponding comparison between the breed at GD49 (Figure 3 and Appendix A). Some of these genes are expressed at higher levels in Meishan sows at GD72 such as *EGR1* [59], *EGF* [55], *ESM1* [60], and *LOXL2* [61], which may enhance endometrial angiogenesis in Meishan pigs. These results potentially explain the denser vascularization of the maternal-fetal interface in Meishan pigs during mid-late gestation (compared to commercial breeds). The same factors may also enhance the uterine receptivity of Meishan pigs. 

Cytokine family members, such as interleukins and interferons, are known to function in porcine pregnancy [62,63]. In this study, we identified common DEGs between Meishan and Duroc at both GD49 and GD72 that are involved in the regulation of type I interferon signaling pathway (*IFI6, IFIT3, IFNA4, MX1, ISG15,* and *USP18*). *ISG15*, interferon (IFN)-stimulated gene 15, plays a vital role in promoting pregnancy maintenance and embryo development [64,65]. Embryo mortality increases in *ISG15*-/- mice [45], and *ISG15* may affect fetal growth, placenta development, and potential infection defense mechanisms in humans [64]. We found that *IGS15* has a higher expression level in Meishan than in Duroc pigs, which may increase the resistance of the Meishan breed to conditions such as maternal hypoxia or crowded uterus.

Estradiol-17β (E_2_β), one of the most important steroid hormones in pregnancy, plays a key role in the high prolificacy of Meishan pigs. Lower levels of E_2_β secretion decrease the trophectoderm mitotic rate early in gestation in Meishan pigs, resulting in a smaller embryo and a higher survival rate [66]. Previous studies showed that E_2_β increases placental size and consequently decreases placental efficiency [67]. Reduced placental efficiency is thought to be the main cause of fetal loss during mid-late gestation in pigs [12]. In our study, enriched GO terms that were in common between DU72 vs. MS72 and MS49 vs. MS72 in the miRNA-mRNA interaction networks included genes and miRNAs involved in responding to estradiol (Figure 6 and Figure 7 and Table 3). Down regulated genes in MS72 such as *IHH* and *ESR1* are known to affect placental development in mammals [68,69,70]. Estradiol regulates expression of *IHH* in the mammalian uterus and plays a role in development of the placenta [68,69]. The effects of estradiol in the uterus are mediated by *ESR1*, which is the predominant type of estrogen receptor in the mature uterus [70,71]. Based on the results of the dual luciferase assay, ssc-miR-671-5p appears to downregulate the expression of *ESR1* (Figure 9B). Lower expression of *IHH* and *ESR1* may be partly responsible for decreasing conceptus and placental size and promoting placental efficiency in Meishan pigs during late gestation. Other genes identified in our analyses (*DUSP1*, *EGFR*, *F7*, *PTCH1*, *SSTR1*, *WFDC1*, *BAD*, and *WNT7A*) may also contribute in important ways to the high placental efficiency of the Meishan breed.

## 5. Conclusions

In summary, RNA-sequencing analysis identified differences in gene and miRNA expression in endometrial tissue from Meishan and Duroc sows during mid-late gestation. miRNA-gene interactions were predicted and used to construct interaction networks. We identified several genes and miRNAs that affect conceptus and placental size by regulating estradiol and angiogenesis at the maternal-fetal interface. These effects may increase placental efficiency and decrease the fetal loss rate in Meishan sows during mid-late gestation. Our findings suggest different regulation patterns in the endometrium of Meishan and Duroc sows and reveal important molecular mechanisms affecting embryonic mortality in pigs during mid-late gestation.

## Figures and Tables

**Figure 1 animals-10-00420-f001:**
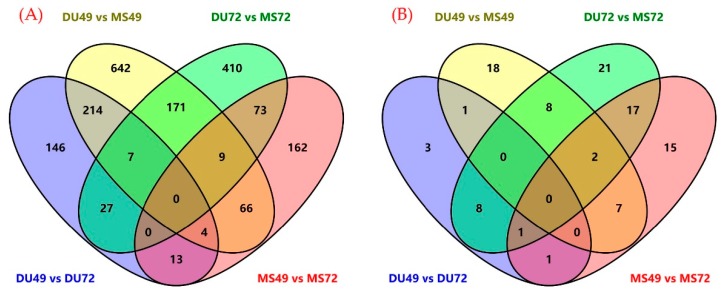
Venn diagram displaying the overlap among different groups of differentially expressed genes (DEGs) (**A**), differentially expressed (DE) miRNAs (**B**).

**Figure 2 animals-10-00420-f002:**
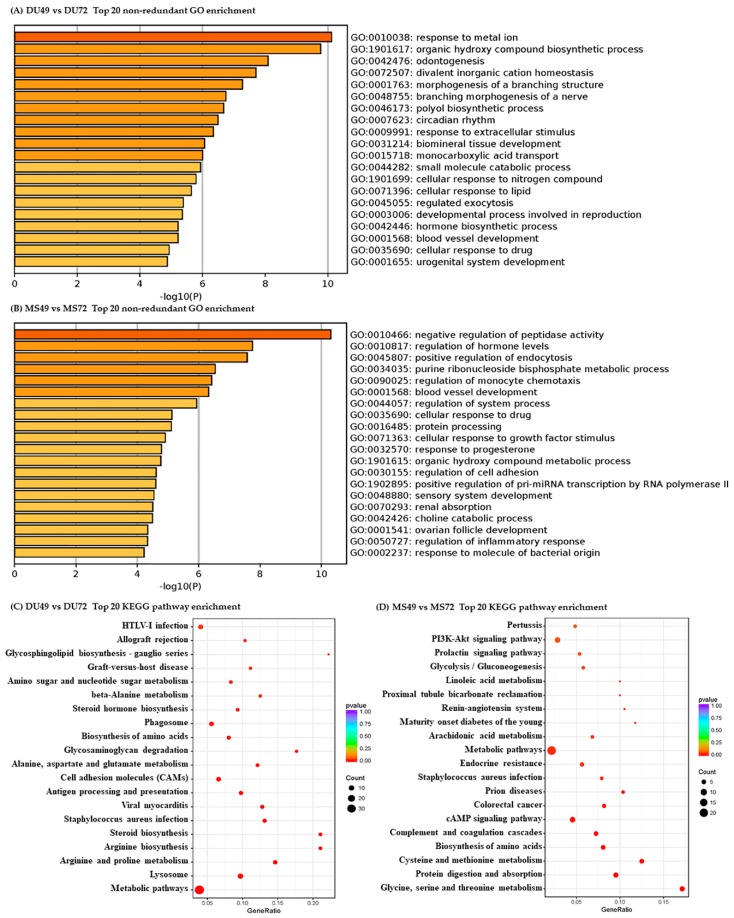
Pathways clustered from DEGs detected between GD49 and GD72. Top 20 non-redundant enriched gene ontology (GO) terms (**A**,**B**) and top 20 enriched KEGG pathways (**C**,**D**).

**Figure 3 animals-10-00420-f003:**
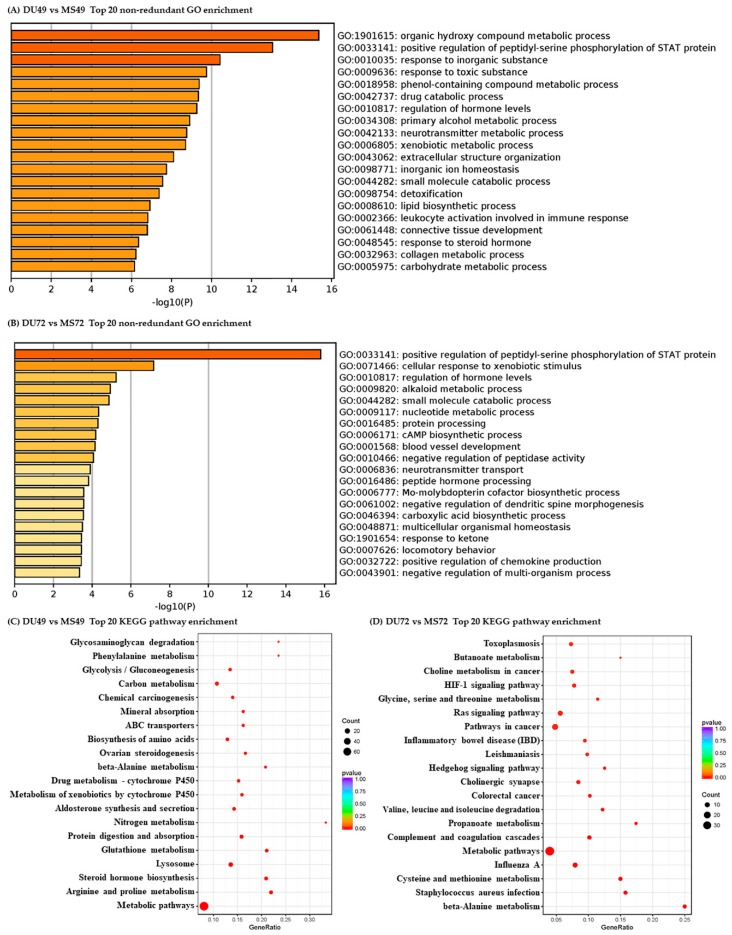
Pathways clustered from DEGs detected between Duroc and Meishan sows. Top 20 non-redundant enriched GO terms (**A**,**B**) and top 20 KEGG pathways (**C**,**D**).

**Figure 4 animals-10-00420-f004:**
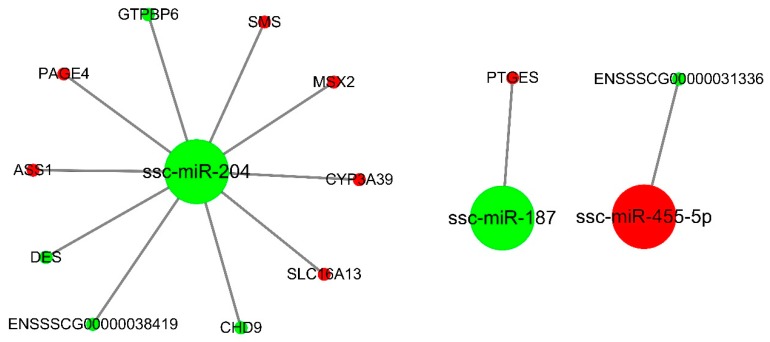
miRNA-gene interaction network (DU49 vs. DU72). Large and small circles represent miRNAs and genes, respectively. Red indicates that the RNA is relatively more abundant, and green indicates that the RNA is relatively less abundant in DU49 than in DU72.

**Figure 5 animals-10-00420-f005:**
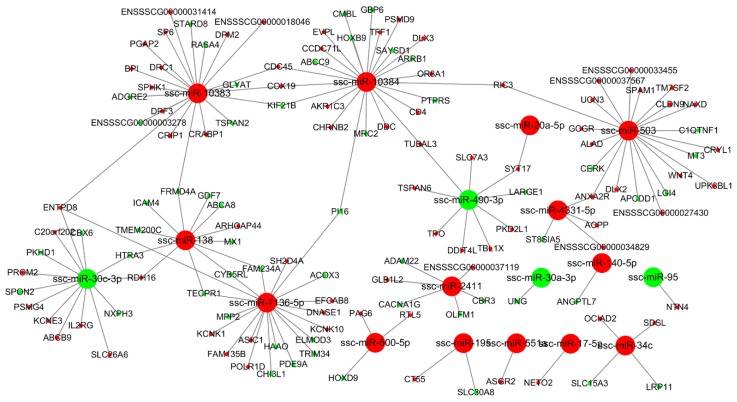
miRNA-gene interaction network (DU49 vs. MS49). Large and small circles represent miRNAs and genes, respectively. Red indicates that the RNA is relatively more abundant, and green indicates that the RNA is relatively less abundant in DU49 than in MS49.

**Figure 6 animals-10-00420-f006:**
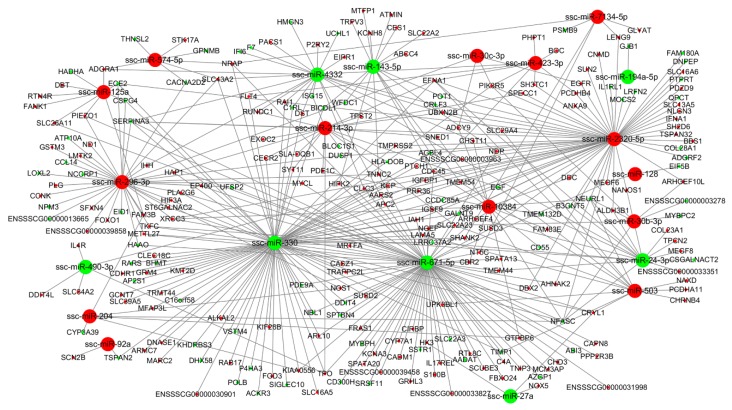
miRNA-gene interaction network (DU72 vs. MS72). Large and small circles represent miRNAs and genes, respectively. Red indicates that the RNA is relatively more abundant, and green indicates that the RNA is relatively less abundant in DU72 than in MS72.

**Figure 7 animals-10-00420-f007:**
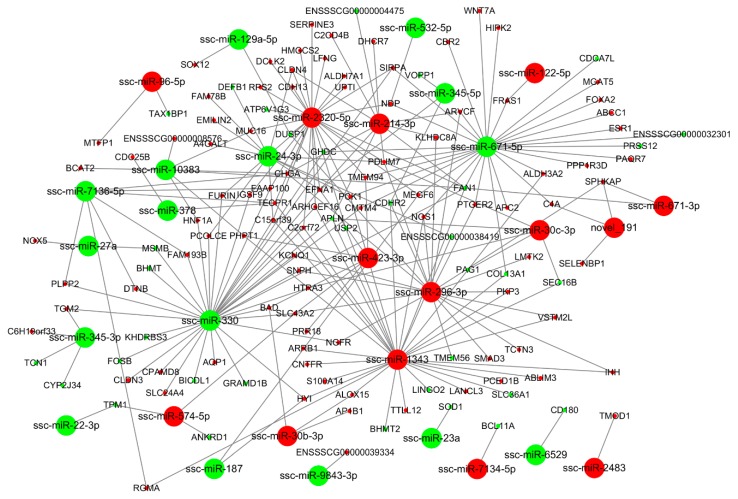
miRNA-gene interaction network (MS49 vs. MS72). Large and small circles represent miRNAs and genes, respectively. Red indicates that the RNA is relatively more abundant, and green indicates that the RNA is relatively less abundant in MS49 than in MS72.

**Figure 8 animals-10-00420-f008:**
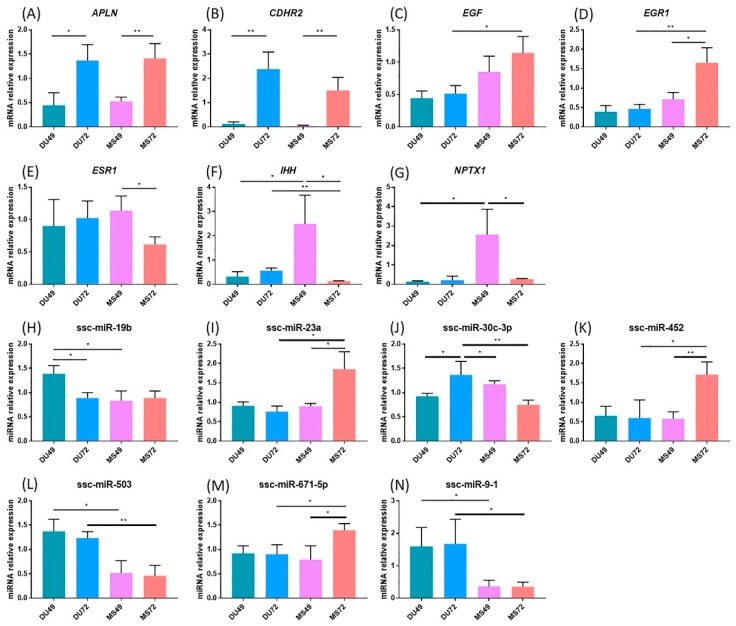
Quantitative PCR validation. The x-axis represents the different stages and breeds and the y-axis shows the fold changes in expression (* *p* < 0.05, ** *p* < 0.01). (**A**) *APLN*; (**B**) *CDHR2*; (**C**) *EGF*; (**D**) *EGR1*; (**E**) *ESR1*; (**F**) *IHH*; (**G**) *NPTX1*; (**H**) ssc-miR-19b; (**I**) ssc-miR-23a; (**J**) ssc-miR-30c-3p; (**K**) ssc-miR-452; (**L**) ssc-miR-503; (**M**) ssc-miR-671-5p; (**N**) ssc-miR-9-1. Relative expression values were normalized as descried in Materials and Methods.

**Figure 9 animals-10-00420-f009:**
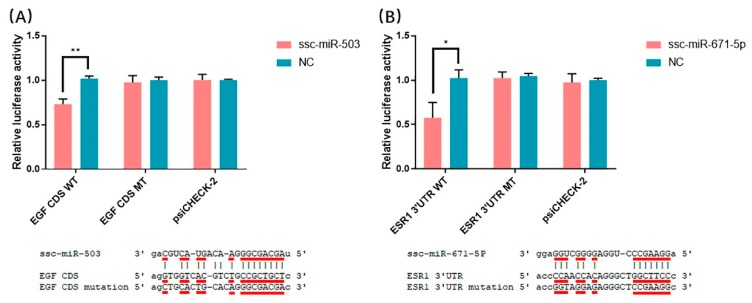
Validation of miRNA-gene interactions. (**A**) Validating *EGF* as a target of ssc-miR-503, (**B**) validating *ESR1* as a target of ssc-miR-671-5p. (* *p* < 0.05, ** *p* < 0.01). WT, wild type; MT, mutant type.

**Table 1 animals-10-00420-t001:** Number of differentially expressed mRNAs and miRNAs.

Comparison	DE mRNAs			DE miRNAs		
	Total	Up	Down	Total	Up	Down
DU49 vs. DU72	411	273	138	14	10	4
DU49 vs. MS49	1113	678	435	36	24	8
DU72 vs. MS72	697	397	300	57	26	31
MS49 vs. MS72	327	185	142	43	21	22

**Table 2 animals-10-00420-t002:** Differentially expressed genes associated with fetal loss or litter size based on previous studies.

Comparison Group	Ensembl ID	Gene Name	Log_2_ Fold Change	Adjusted *p*-Value
DU49 vs. DU72	ENSSSCG00000030484	*AHR* [34]	−0.62	0.0247
	ENSSSCG00000004283	*DPPA5* [35]	4.68	0.0098
	ENSSSCG00000010698	*FGFR2* [36]	0.96	0.0446
	ENSSSCG00000012229	*GP91-PHOX* [37]	−0.74	0.0467
	ENSSSCG00000015579	*PTGS2* [38]	1.85	0.0031
	ENSSSCG00000037754	*SLC39A11* [39]	1.08	0.0235
DU49 vs. MS49	ENSSSCG00000013303	*ABTB2* [40]	−0.98	0.0497
	ENSSSCG00000017101	*ADCY2* [41]	−1.35	0.0137
	ENSSSCG00000011437	*ALAS1* [42]	1.33	0.0283
	ENSSSCG00000025578	*ALDH1A2* [43]	−1.24	0.0422
	ENSSSCG00000004283	*DPPA5*	3.85	3.66 × 10^−^^6^
	ENSSSCG00000012229	*GP91-PHOX*	−1.05	0.0174
	ENSSSCG00000016204	*IHH* [44]	−1.50	0.0049
	ENSSSCG00000040575	*ISG15* [45]	−1.23	0.0171
	ENSSSCG00000015579	*PTGS2*	1.87	0.0419
	ENSSSCG00000007836	*SCNN1G* [38]	1.44	0.0412
DU72 vs. MS72	ENSSSCG00000009134	*EGF* [46]	−1.57	0.0012
	ENSSSCG00000014336	*EGR1* [47]	−1.44	3.75 × 10^−^^6^
	ENSSSCG00000002383	*FOS* [47]	−0.79	0.0040
	ENSSSCG00000010639	*HABP2* [41]	2.48	0.0248
	ENSSSCG00000016204	*IHH*	2.20	5.36 × 10^−^^8^
	ENSSSCG00000040575	*ISG15*	−1.01	0.0059
	ENSSSCG00000000455	*LRIG3* [48]	2.14	1.37 × 10^−^^10^
MS49 vs. MS72	ENSSSCG00000014336	*EGR1*	−1.59	0.0002
	ENSSSCG00000025777	*ESR1* [49]	0.85	0.0036
	ENSSSCG00000002383	*FOS*	−1.34	9.03 × 10^−^^8^
	ENSSSCG00000015144	*GRAMD1B* [40]	−1.39	0.0294
	ENSSSCG00000016204	*IHH*	4.13	6.46 × 10^−^^19^

**Table 3 animals-10-00420-t003:** Significantly enriched GO terms and associated KEGG pathways shared between DU72 vs. MS72 and MS49 vs. MS72 in miRNA-mRNA interaction networks.

Comparison Group	Terms	DEGs No.	*p*-Value	Genes
DU72 vs. MS72	GO:0032355 ~ Response to estradiol	7	0.00025	*DUSP1, EGFR, F7, IHH, PTCH1, SSTR1, WFDC1*
	GO:0070482 ~ Response to oxygen levels	10	0.00379	*F7, FOXO1, LOXL2, NOS1, POLB, PSMB9, S100B, HIPK2, DDIT4, HIF3A*
	GO:0001525 ~ Angiogenesis	13	0.00337	*CSPG4, EFNA1, EGF, FLT4, IHH, LOXL2, GPNMB, CNMD, HIPK2, ACKR3, HIF3A, NOX5, COL23A1*
	GO:0001568 ~ Blood vessel development	15	0.00497	*CSPG4, EFNA1, EGF, MEGF8, FOXO1, FLT4, IHH, LOXL2, GPNMB, CNMD, HIPK2, ACKR3, HIF3A, NOX5, COL23A1*
	ssc04015 ~ Rap1 signaling pathway	4	3.52 × 10^−^^5^	*FLT4, EGF, EGFR, EFNA1*
	ssc04014 ~ Ras signaling pathway	4	5.41 × 10^−^^5^	*FLT4, EGF, EGFR, EFNA1*
	ssc04151 ~ PI3K-Akt signaling pathway	3	0.00019	*FLT4, EGF, EGFR, EFNA1*
	ssc04068 ~ FoxO signaling pathway	4	0.00031	*EGF, EGFR, FOXO1*
MS49 vs. MS72	GO:0032355 ~ Response to estradiol	5	0.00037	*BAD, DUSP1, ESR1, IHH, WNT7A*
	GO:0070482 ~ Response to oxygen levels	8	0.00045	*AQP1, BAD, CLDN3, SMAD3, NOS1, PCK1, ANKRD1, HIPK2*
	GO:0001525 ~ Angiogenesis	8	0.00550	*AQP1, CDH13, EFNA1, IHH, NGFR, WNT7A, HIPK2, NOX5*
	GO:0001568 ~ Blood vessel development	9	0.00809	*AQP1, CDH13, DHCR7, EFNA1, IHH, NGFR, WNT7A, HIPK2, NOX5*
	ssc04015 ~ Rap1 signaling pathway	2	0.00580	*EFNA1, NGFR*
	ssc04014 ~ Ras signaling pathway	3	0.00026	*BAD, EFNA1, NGFR*
	ssc04151 ~ PI3K-Akt signaling pathway	4	2.26 × 10^−^^5^	*BAD, EFNA1, PCK1, NGFR*
	ssc04068 ~ FoxO signaling pathway	2	0.00291	*PCK1, SMAD3*

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
