# Peer review of "Integrated Analysis of miRNA-mRNA Network Reveals Different Regulatory Patterns in the Endometrium of Meishan and Duroc Sows during Mid-Late Gestation"

_animals, 2020, doi:10.3390/ani10030420_

Round 1
Reviewer 1 Report
The Authors added the information about the animals used in the study and explained how the mRNAs and microRNAs were selected for qPCR validation. I accept that. If the manuscript has been checked and corrected for English spelling, grammar, and readability by a professional editing service, I don't feel qualified to judge further about the English.
I've asked to improve the quality of the figures 5, 6 and 7. The names of the genes were hardly readable or unreadable.
In the cover latter Authors claim that “Figures 5-7 have been redrawn with larger fonts. In addition, the predicted miRNA-gene pairs are now listed in Table S12”. However in the corrected version of the manuscript, the Authors have corrected only the figure 7 (it is ok now). The figures 5 and 6 are still unreadable. Did the Authors uploaded the wrong version of the corrected manuscript (missing the corrections of Figures 5 and 6)? They also wrote, that they have added the table S12 but I don’t see such a table in supplementary files. Such a table would be helpful for the reader. Where is it?
Apart from that, the figures 2 and 3 are also poor quality - they are readable, so I haven’t mention that before, but the resolution is low and figures are not clear enough. Figures 2 and 3 should be also improved.
Placing the hardly readable or unreadable figures with low resolution and visible pixels is incomprehensible. If the reader can’t read the figures, the figures are redundant and useless. The figures 2, 3, 5, 6 have to be improved. The figures presentation is the weak point of the manuscript, but it can be easily corrected. That not discredit the article, but make the article inconvenient to read.
Author Response
The Authors added the information about the animals used in the study and explained how the mRNAs and microRNAs were selected for qPCR validation. I accept that. If the manuscript has been checked and corrected for English spelling, grammar, and readability by a professional editing service, I don't feel qualified to judge further about the English.
Response: Thank you.
I've asked to improve the quality of the figures 5, 6 and 7. The names of the genes were hardly readable or unreadable.
In the cover latter Authors claim that “Figures 5-7 have been redrawn with larger fonts. In addition, the predicted miRNA-gene pairs are now listed in Table S12”. However, in the corrected version of the manuscript, the Authors have corrected only the figure 7 (it is ok now). The figures 5 and 6 are still unreadable. Did the Authors uploaded the wrong version of the corrected manuscript (missing the corrections of Figures 5 and 6)? They also wrote, that they have added the table S12 but I don’t see such a table in supplementary files. Such a table would be helpful for the reader. Where is it?
Response: There is something wrong with the last version. Figures 5-6 have been redrawn with larger fonts. The predicted miRNA-gene pairs are now listed in Table S12.
Apart from that, the figures 2 and 3 are also poor quality - they are readable, so I haven’t mention that before, but the resolution is low and figures are not clear enough. Figures 2 and 3 should be also improved.
Placing the hardly readable or unreadable figures with low resolution and visible pixels is incomprehensible. If the reader can’t read the figures, the figures are redundant and useless. The figures 2, 3, 5, 6 have to be improved. The figures presentation is the weak point of the manuscript, but it can be easily corrected. That not discredit the article, but make the article inconvenient to read.
Response: Thank you for these suggestions. Figures 2-3 have been redrawn with larger fonts.
Reviewer 2 Report
The authors have included some of the comments raised in the first review but they have not made major changes in the quality of the writing style. Text editing is needed.
Author Response
The authors have included some of the comments raised in the first review but they have not made major changes in the quality of the writing style. Text editing is needed.
Response: The revised manuscript has been thoroughly edited for English by a professional editing service. Please see the details.
This manuscript is a resubmission of an earlier submission. The following is a list of the peer review reports and author responses from that submission.
Round 1
Reviewer 1 Report
Yang et al made integrated analysis of miRNA-mRNA network which reveals the different regulatory patterns in the endometrium during the mid-late gestation from Meishan and Duroc sows. Their findings might provide an important resource for studying embryonic mortality during mid-late gestation in pigs. There are some minor points that needed to be clarified.
More information about the chosen pigs should be supplied. Such as age, parity.
The part from Line 225 to line 229 was a repetition of the part from line 204 to 208.
Reviewer 2 Report
The manuscript "Integrated Analysis of miRNA-mRNA Network 2 Reveals the Different Regulatory Patterns in the 3 Endometrium During the Mid-Late Gestation from 4 Meishan and Duroc Sows" offers the results from a well-designed study for comparing findings in two swine breeds differing in prolificacy due to main differences in embryo/fetal survival.
The main flaw of the work are related to the quality of the writing English in the manuscript. I think that the text would be benefited by a revision by an English speaking colleague. Furthermore, I think that the discussion section should include a deeper consideration of the physiological significance of the findings; mainly when related to estradiol, which is largely known as the critical point for the higher prolificacy of the Meishan breed. Finally, the conclusion is really poor; authors should include more relevant information on the data obtained in the study.
Reviewer 3 Report
The manuscript focuses on differential expression of genes and miRNA at mid-late gestation in Meishan and Duroc pig breeds. The article provide new information and could be useful for future studies and for revealing the molecular mechanisms of embryonic mortality. The topic should be important in the field of pig breeding. It will be interesting for the readers.
In general, the language of the manuscript is understandable, but I am not the English native speaker. I found some confusing phrases, which probably should be checked by the native speaker. For example: in the line 54, authors write: “Meishan embryos have less development (…)” or the mistake in the line 136: “Micor RNA Sequencing”.
The title comprehensively describes the article content.
The abstract is well constructed.
The methods were properly chosen and well described. However it is not clear whether animals were kept in similar housing (same piggery?) and feed conditions, what sometimes may influence the gene expression results. Did pigs were slaughtered in the same day/week/month?
The results are presented in three tables, nine figures and supplementary materials and support the conclusion that some of the genes and miRNAs are involved in important biological processes associated with the development process of endometrium and provide the information on differentially expressed genes and miRNAs between Meishan and Duroc sows during the mid-late gestation.
The figures 5-7 are hardly readable (magnifying the page does not help – at least in my copy of the manuscript). The fonts should be sharper or bigger. I understand that those figures are made by the software which has some limitations but I my opinion the miRNA and genes written names could be improved for the publication.
In lines 328-331: Author write that “(…)seven differentially expressed mRNAs (Table S8) and seven 328 differentially expressed microRNAs (Table S9) were selected (…)”. How the mRNAs and microRNAs were selected for qPCR validation?
The discussion is well prepared. The references are well selected.
The following comment is meant to be helpful to the Authors:
It is a pity that the authors were not tempted to study placental vascularization and estrogen in pigs used in the experiment. This could strengthen the discussed conclusions and suggestions. This remark does not discredit the manuscript, but it should help the Authors in their future studies.
In conclusion, I think that the manuscript should be published in Animals.
